# Determinants of acceptability of schistosomiasis mass drug administration among primary school children in Busega District, Northwestern Tanzania

Irene Yunzu Edward [1,2]*, Patricia Maritim[3], Choolwe Jacobs[2], Adam Silumbwe[3], Hussein Mohamed[4], Joseph Mumba Zulu[3], Hikabasa Halwiindi[5]

1 Department of Medical Sciences and Technology, College of Science and Technical Education, Mbeya University of Science and Technology, Mbeya, Tanzania, 2 Department of Epidemiology and Biostatistics, School of Public Health, University of Zambia, Lusaka, Zambia, 3 Department of Health Policy and Management, School of Public Health, University of Zambia, Lusaka, Zambia, 4 Department of Environmental and Occupational Health, School of Public Health and Social Sciences, Muhimbili University of Health and Allied Sciences, Dar es salaam, Tanzania, 5 Department of Community and Family Medicine, School of Public Health, University of Zambia, Lusaka, Zambia

* ireneyunzu@gmail.com

## Abstract

### Background

Schistosomiasis is endemic in Tanzania, with a prevalence ranging between 12.7% to 87.6%. Mass drug administration (MDA) with praziquantel is the main recommended choice of preventive treatment for the disease. Low treatment coverage rates (46.6%) in Busega District, Northwestern Tanzania which are far less than the recommended effective coverage rates of ≥ 75% could indicate low acceptability and poor uptake of MDA. This study sought to establish factors associated with acceptability of schisto-somiasis mass drug administration among primary school children in Busega District.

### Methods

A cross-sectional study was conducted among primary school children, randomly selected from six primary schools between 6th March and 29th May 2023. A validated generic questionnaire guided by the Theoretical Framework of Acceptability was used for data collection. The outcome of the study was acceptability of MDA, and the explanatory variables were socio-demographic factors and the seven constructs of the framework such as perceived moral about taking praziquantel, understanding of intervention purpose and affective attitude. Descriptive statistics and logistic regression with robust standard errors were conducted to identify factors associated with acceptability of MDA using STATA version 15.

**Data availability statement:** All relevant data are in the Supporting Information file.

**Funding:** The author(s) received no specific funding for this work.

**Competing interests:** The authors have declared that no competing interests exist.

## Results

The study sample comprised 615 primary school children, 60.16% girls and 39.84% boys. Age distribution ranged from 10 to 17 years with a median age of 13 years. About 55.28% were found to have high acceptability of MDA. Factors significantly associated with acceptability of MDA were; perceived effectiveness (AOR = 2.52; 95%CI = 1.31–4.85; p-value = 0.006), understanding of intervention purpose (AOR = 5.51; 95%CI = 3.16–9.59; p-value<0.0001), self-efficacy (AOR = 2.04; 95% CI = 1.08–3.85; p-value = 0.029), affective attitude (AOR = 5.10; 95% CI = 2.77–9.59; p-value<0.0001), and gender (AOR = 0.59; 95% CI = 0.38–0.94; p-value = 0.027).

## Conclusion

Slightly more than half of primary school children recorded high acceptability of MDA. However, perceived effectiveness, understanding of intervention purpose, self-efficacy, affective attitude, and gender strongly influence acceptability. This underscores the need for tailored, gender-sensitive community sensitization efforts regarding the benefits of MDA. Targeted educational campaigns and peer-led initiatives should be prioritized to increase awareness and acceptance, ultimately improving the effectiveness of the MDA program.

## Introduction

Globally, more than 250 million people are infected with schistosomiasis [1]. The disease is generally common in low-income communities without access to potable water, proper environmental sanitation and hygiene [2]. Transmission occurs when infected people contaminate water bodies with excreta containing eggs of the – parasite, which hatch in the water [3]. Schistosomiasis can lead to hepatomegaly, splenomegaly, anemia, kidney malfunction, stunting growth in children and reduce ability to learn in school/cognitive dysfunction in children [4]. Approximately 93% of schistosomiasis cases are concentrated in sub-Saharan Africa (SSA), with Tanzania ranking as the second country with the highest prevalence of the disease [5].

In Tanzania, the prevalence of the diseases ranges between 12.7% to 87.6% [6]. An estimated 43.5 million people remains at risk of the disease and the prevalence appears to heighten as the population size grows [7]. Busega District has documented a concerning prevalence of intestinal schistosomiasis among school-aged children, reaching as high as 90.6% [8].

In the absence of a safe and effective vaccine, Praziquantel is the current recommended choice of preventive treatment for schistosomiasis [3]. Praziquantel MDA in Tanzania was introduced in 2004 and its mode of delivery includes both school-based approach for all school aged children and community-based approach for the highly infected communities [9]. The Ministry of Health and other development partners including the Interchurch Medical Assistance (IMA), Schistosomiasis Control Initiative (SCI, currently known as Unlimited Health), World Health Organization

(WHO) and Research Triangle Institute (RTI) International forms the main implementers of the MDA program in the country. The MDA delivery in schools is conducted by trained schoolteachers supported by healthcare personnel in nearby health facilities. Between 2005–2016, schistosomiasis control implementation was scaled up to 26 regions covering approximately 4 million school-aged children by 2020 [6]. Available data indicate that the program has now achieved 100% geographical coverage, suggesting that the reach of MDA is good. Nevertheless, low epidemiological coverage in some districts and sub districts remains a common challenge [6]. For instance, a study conducted in Bahi district within Dodoma region of Tanzania found that MDA coverage was only 43.6% [10]. Relatedly, another study conducted in the urban setting- Morogoro region of Tanzania found that, of the 884 sampled students, only 126 (14.3%) took praziquantel in the last annual MDA campaign [11]. Similarly, a study conducted in North-western parts of Tanzania revealed that only 12.8% of primary school students participated at least once in the MDA program [1]. Arguably, the findings from these studies indicate the need for sustained efforts to enhance MDA uptake in the country, thus facilitating more effective disease control measures.

Ongoing control efforts by the National Neglected Diseases Control Plan include provision of at least once-a-year praziquantel treatment to school aged children in the affected communities as well as community sensitization efforts with an emphasis on disease prevention [12]. Despite these commitments, the 2021 treatment records suggest that Busega District had treatment coverage of 46.6%, far less than the global target of ≥ 75% in a given population [3]. Low coverage could mean lack of acceptability, possibly resulting in poor uptake of MDA. Acceptability is the perception among implementation stakeholders that a given treatment, service, practice or innovation is agreeable, palatable, or satisfactory [13]. A previous review has suggested that if an intervention is considered acceptable patients are more likely to adhere to the treatment [14]. Furthermore, a study in North-western Tanzania highlighted that important challenge of the MDA campaign is acceptance among the targeted population towards the treatment [15]. Low treatment acceptability has been shown to have an impact on overall effectiveness of the intervention and failure to achieve the disease control [16].

Previous studies have examined the factors associated with uptake/ low coverage of praziquantel MDA in other countries [17–19]. However, studies in Tanzania have primarily focused on the impact of praziquantel MDA campaign on prevalence and intensity of the infection among school aged children, for example [10,11,20–22]. Little is known in Tanzania on the acceptability of MDA among school children which create a gap that this study seeks to address. Given the localized nature of the disease, there is a pressing necessity for targeted investigations within this specific setting to provide actionable insights for future intervention strategies. Therefore, this study assessed factors associated with acceptability of schistosomiasis MDA among school children in Busega District, North-western Tanzania.

## Materials and methods

### Theoretical framework

This study was guided by the theoretical framework for acceptability (TFA) [14]. The TFA comprises seven domains including *Perceived effectiveness, Intervention coherence, Affective attitude, Self-efficacy, Opportunity cost, Burden*, and *Ethicality*. Perceived effectiveness is conceived as the extent to which MDA is perceived as likely to achieve its purpose. Intervention coherence is defined as the extent to which participants understand the purpose of the intervention (MDA) and how it works. Affective attitude refers to how individuals feel about taking praziquantel. Additionally, self-efficacy denotes the participant's confidence that they can perform the behavior(s) required to participate in schistosomiasis MDA. Opportunity cost refers to the benefits, profits or values that must be given up receiving schistosomiasis MDA. Burden implies the perceived number of efforts that is required to participate in receiving MDA. Finally, ethicality symbolizes the extent to which the intervention has good fit with an individual's values system.

The choice of TFA was driven by the focus on acceptability. Also, it has been used in previous studies [23,24].

TFA guided the whole research process including study tool design, data collection, data analysis, and the interpretation of findings.

## Study design

A school-based cross-sectional study was conducted to determine the acceptability of schistosomiasis MDA among primary school children and its associated factors in Busega District.

## Study setting

The study was conducted between 6th March 2023 and 29th May 2023 in Busega District, Tanzania. The last round, which MDA was conducted in this district was 2021. No mass praziquantel treatment was administered in 2022, a year before data collection for this study was done. Busega district is one of the six districts in Simiyu Region. The district's population is about 282,167 according to Tanzania national population and housing census survey of 2022. It covers a total area of 1,424km$^2$ with thirteen (13) wards. Busega District is bordered to the north by Lake Victoria and Bunda District, to the east by Bariadi District, and to the south by Magu District. The district receives two rounds of rainfall per year, light rains around October to December and heavy rains around March to May of every year, ranging from 700 mm to 1000 mm. The mean temperature in the district ranges between 18°C to 20°C during rainy season and up to 32°C during dry season.

The main economic activities are farming and fishing in Lake Victoria. Schistosomiasis is among the top ten causes of morbidity and mortality in the district [8]. Transmission of schistosomiasis is facilitated by the presence of freshwater bodies. People are at risk of acquiring infections with *S. mansoni* and S. *haematobium* due to fishing, agricultural, domestic and recreational activities which expose them to water bodies potentially infested with cercariae shedding snail's species. Busega District council has four types of water sources used in the community which include lakes, boreholes, shallow wells, and rainwater. Water coverage in the district is 39.5% [25]. Lack of hygiene in urine, feces disposal and swimming habits make primary school children especially at risk to infection

## Study population

MDA in Busega District is school-based, targeting primary school children. To capture the level of acceptability at the individual level, the study focused on this group. Primary school children are in grades I to VII, and the district had received eight rounds of MDA (2005, 2007, 2013, 2014, 2015, 2018, 2019, and 2021). School children in grades V–VII had the full benefit of two to three rounds of MDA (2018, 2019, and 2021). Therefore, this survey focused on schoolchildren in grades V–VII. The decision to include only students in grades V–VII was guided by several considerations, including the reliability and depth of data collected. Older students are more likely to comprehend and articulate their experiences, perceptions, and concerns regarding the intervention, which is critical in assessing acceptability. This approach also aligns with similar studies on MDA, which have prioritized upper primary students [26,27].

Eligible respondents were primary school children living in Busega district, from standard five up to seven, whose parents/guardians gave informed consent for their participation, and who provided assent to participate were included in the study.

## Sampling and sample size

The study sample size was estimated using the following formula $n = Z^2 p (1-p)/\varepsilon^2 \times f$ whereby, n = the minimum estimated sample size; Z = standard normal deviate (1.96 for 95% confidence interval); p = expected proportion, (0.413) from previous studies in Northwestern, Tanzania [25]. ε is the margin of error, settled at 5%; f is a correction for design effect of 1.5 and 10% of anticipated non-response rate was also applied.

According to sample size calculation, a total of 615 primary school children were enrolled by the multi-stage cluster sampling technique. In the first stage Busega District was stratified according to geographical location of the primary schools: those that are nearby Lake Victoria (within a 10 km distance from Lake Victoria) and those far from Lake Victoria

(more than 10 km distance from Lake Victoria). The reference of 10 Km was arrived at based on another study [28]. In the second stage of selection, a simple random method was used to select 3 primary schools, from each stratum. A total of six primary schools (Itongo, Chamugasa, Mkula, Malangale, Lutubiga, and Kabita primary school) were selected. Then, in the third and final stage, a systematic random selection of eligible primary school children from each school was conducted. Specifically, 103 participants each from Kabita, Malangale, Lutubiga, Itongo, chamugasa and Mkula primary school, resulting in a total sample size of 615 primary school children.

## Data collection

A validated generic questionnaire guided by theoretical framework of acceptability was used for the data collection [29]. Additional questions, such as those related to socio-demographic characteristics and the acceptability of the intervention measures, were included to capture all information related to individuals. During data collection, the questionnaire was translated into Swahili and then created in online software (ODK) with closed-ended questions. Data were collected using mobile phones or tablets with installed ODK software, according to a data plan schedule. All interviews were conducted face-to-face by trained researcher assistants and the first author.

## Variable description and measurements

The outcome variable, acceptability of MDA, was assessed using acceptability of intervention measures such as "intervention meets my approval," "intervention is appealing to me," "I like the intervention," and "I welcome the intervention." Six questions were prepared in line with another study [30] using a 5-point Likert scale (1–5) to assess participant responses. The variable was then dichotomized as low acceptability for participants with scores below the median (50th percentile) and high acceptability for those with scores equal to or above the median (coded as 0 = low acceptability level and 1 = high acceptability level), aligns with the study conducted in Ghana [31].

The socio- demographic variables included: Age which was measured as continuous; gender, defined as boy and girl; class of study, categorized as standard V, VI and VII; parents level of education, classified as don't know, no education, primary, secondary and tertiary; religion, grouped as no religion, Christian, Muslim and geographical location of the school defined as nearby lake and far from the Lake Victoria.

Other variables included affective attitude, perceived effectiveness, self-efficacy, interference with school activities, understanding of intervention purpose, and perceived moral about taking praziquantel. These variables were collected using a 5-point Likert rating scale ranging from 1 'Strongly disagree' to 5 'Strongly agree,' following the generic TFA questionnaire [29]. For analysis purposes, these variables were dichotomized. A score above the median ($>3$) was coded as 1 'Yes,' while a score less than or equal to the median ($\leq 3$) was coded as 0 'No' for perceived effectiveness, interference with school activities, understanding of intervention purpose, and perceived moral about taking praziquantel. Additionally, for self-efficacy, 1 'Confident' and 0 'Unconfident,' and for affective attitude, 1 'Comfortable' and 0 'Uncomfortable.' This categorization was adopted from another study [18].

## Data management and analysis

The primary data collected were imported, cleaned, coded and analyzed using the statistical software STATA version 15 (STATA corp. college station, Taxes, US). The Cronbach Alpha which measures the internal consistency (the extent to which the items in a test measure the same component) of a scale was used. A calculated alpha value of 0.7851 which is closer to one indicates that the item measures the component well. Descriptive statistical analysis was performed first to observe the characteristics of the variables. The main statistical analysis consisted of univariable and multivariable logistic regression with robust standard errors to identify factors associated with acceptability of schistosomiasis MDA. An investigator led backward stepwise logistic regression was used to arrive at the model that explains well the data (best fit model)

The selection of variables that fit in the final multiple regression model (best fit model) was done by running the multiple logistic regression command with all the variables that were significant at bivariable analysis. Then, those with the highest p-values were removed one by one from the model until only significant variables that best predict the outcome remained. Finally, the best fit model was selected based on the Akaike's Information Criterion and Bayesian Information Criterion (AIC and BIC) for the competing models. The model with the smallest values for AIC and BIC compared to another model was chosen. Crude (cOR) and the adjusted odds ratios (aOR) with their corresponding 95 percent confidence intervals (CI) were presented.

## Ethical considerations

Ethical approval to conduct this study was sought from both University of Zambia Biomedical Research Ethics Committee (UNZABREC) with REF. No.3454−2022 and Lake Zone Institutional Review Board (LZIRB), National Institute for Medical Research (NIMR) – Mwanza, Tanzania approval number (MR/53/100/730). Permission to collect data was obtained from District medical officer (DMO), Ward executive director (WEO), and school head teachers. Written informed consent was acquired from teachers on behalf of the parents who verbally consented for their children to participate in the study. The teachers witnessed the verbal consent process, and the researchers documented the names of the parents who provided consent. This ensured that only children whose parents gave consent were included in the study. In addition, assent forms were provided to eligible primary school children. The consent process was conducted individually, whereby each child who agreed to participate signed an assent form. The participation in the study was voluntary, and participants had the right to refuse or withdraw from the study at any point without facing any negative consequences.

## Inclusivity in global research

Additional information regarding the ethical, cultural, and scientific considerations specific to inclusivity in global research is included in the Supporting Information (S3 File).

## Results

### Socio- demographic characteristics of the study participants

In this study, a total of 615 primary school children were interviewed, with a response rate of 100%. Approximately 60.16% (370/615) were girls, while 39.84% (243/615) were boys, ranging from standard V to VII. Their age was not normally distributed, with a median age and interquartile range of 13 (12–14) years. Furthermore, the majority of participants were Christian, accounting for 93.82% (577/615). Moreover, most of their parents had completed primary education, comprising 59.02% (363/615), while 9.43% (58/615) had no education (Table 1).

### Acceptability of schistosomiasis MDA among primary school children

Of the 615 primary school children who were interviewed, about 55.28% (340/615) scored high acceptability while 44.72% (275/615) scored low acceptability. (Fig 1)

### Factors associated with acceptability of schistosomiasis MDA among primary school from the best fit model

The primary school children who understood the purpose of MDA were 6 times more likely to have high acceptability compared to those who were unaware of its purpose (AOR = 5.51; CI = 3.16–9.59; p-value < 0.0001). A positive association was observed between primary school children who felt comfortable taking schistosomiasis drugs and high acceptability of schistosomiasis MDA compared to those who felt uncomfortable (AOR = 5.10; 95% CI = 2.77–9.59; p-value <0.0001). Those primary school children who perceived schistosomiasis drugs as effective were 3 times more likely to have high acceptability of schistosomiasis MDA compared to their counterparts (AOR = 2.52; 95% CI = 1.31–4.85; p-value = 0.006).

**Table 1. Social demographic characteristics of primary school children.**

| Characteristics | Frequency (n = 615) | Percentage (%) |
|---|---|---|
| Age (in years) | IQR | 13(12,14) |
| Gender | | |
| Boys | 245 | 39.84 |
| Girls | 370 | 60.16 |
| Class | | |
| Standard V | 33 | 5.37 |
| Standard VI | 338 | 54.96 |
| Standard VII | 244 | 39.67 |
| Parents level of education | | |
| Don't know | 74 | 12.03 |
| No education | 58 | 9.43 |
| Primary | 363 | 59.02 |
| Secondary | 98 | 15.93 |
| Tertiary | 22 | 3.58 |
| Religion | | |
| Christian | 577 | 93.82 |
| Muslim | 11 | 1.79 |
| No religion | 27 | 4.39 |
| Have ever swallowed praziquantel | | |
| No | 103 | 16.75 |
| Yes | 512 | 83.25 |
| Location of the school | | |
| Nearby lake | 308 | 50.08 |
| Far from the lake | 307 | 49.92 |

IQR = Interquartile range.

Moreover, primary school children who were confident about taking schistosomiasis drugs were 2 times more likely to have high acceptability of schistosomiasis MDA compared to those who were unconfident (AOR = 2.04; 95% CI = 1.08–3.85; p-value = 0.029). Additionally, boys were 41% less likely to have high acceptability of schistosomiasis MDA compared to girls (AOR = 0.59; 95% CI = 0.38–0.94; p-value = 0.027). Nevertheless, age, waiting time, perceived moral about taking praziquantel and interference with school activities were not significantly associated with acceptability of MDA (Table 2).

## Discussion

This study was conducted to determine factors associated with the acceptability of schistosomiasis MDA among primary school children in Busega district. We found that, slightly more than half of the primary school children (55.28%) scored high acceptability of schistosomiasis MDA. The factors associated with acceptability were perceived effectiveness, affective attitude, self-efficacy, understanding of intervention purpose, and gender. Putting this finding into context, records shows that treatment coverage for the last round in the district was 46.6%, far less than the global target [3]. This could mean that an acceptability rate of 55.28% may not be sufficient to achieve the desired treatment coverage. This finding aligns with a study conducted in Ijinga, Tanzania, which reported that less than 50% of the school children demonstrated a high intention of participating in treatment campaigns [15]. Additionally, qualitative findings from Uganda and Ogutu State, Nigeria, further support this result, indicating that poor acceptability was the major barrier to uptake and adequate coverage [32]. On the other hand, these findings contrast with other studies conducted among pre-schoolers (< 6 years)

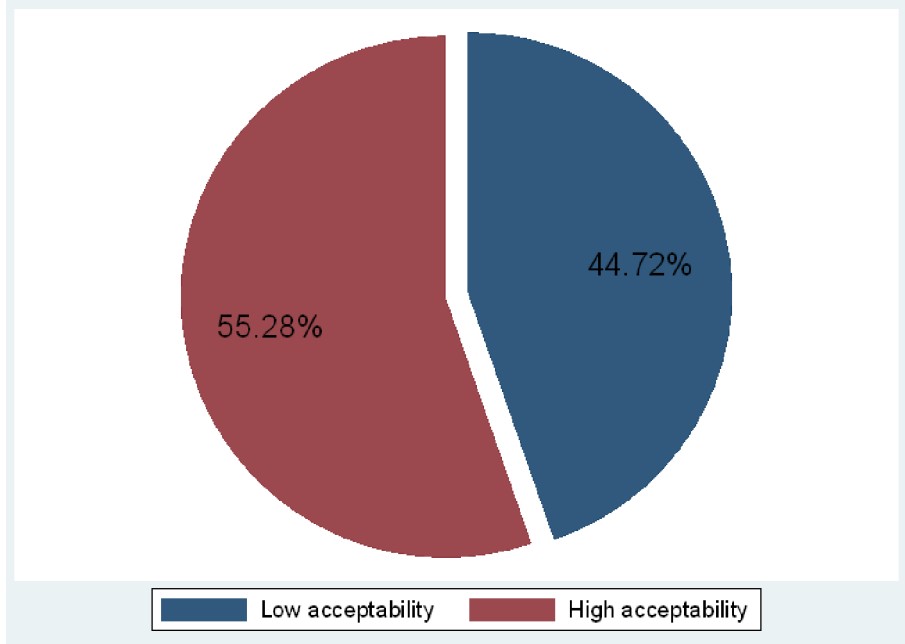

**Fig 1. Distribution of acceptability of schistosomiasis MDA among primary school children.** This figure illustrates the proportion of primary school children who reported high versus low acceptability of MDA.

in Sudan and Kwale, Kenya. These studies found a high acceptability of praziquantel among pre-schoolers, with rates of 99.5% (187/188) [33] and 100% (400/400) [34], respectively. These findings indicate that the acceptability of MDA varies across settings and age groups, highlighting the need for tailored MDA campaign strategies to address specific contexts.

The study revealed that perceived effectiveness was positively associated with acceptability of MDA. A possible explanation for this is that when people perceive a treatment as effective, they may feel more confident in its ability to improve their health and prevent disease. This confidence can lead to a greater willingness to comply with the treatment. The perception of effectiveness could stem from various factors such as prior education about the disease, personal experiences, or information from trusted sources like healthcare providers or community leaders. Same arguments were made in another study [15] which indicated that the perceived effectiveness of praziquantel in curing the disease significantly influenced individuals' willingness to participate in MDA treatment campaigns. Similarly, Spencer *et al* [35] also reported a congruent result, showing that a lack of perceived benefits, particularly in the absence of adequate education, was associated with decreased participation in MDA programs. This finding underscores the importance of conveying clear and accurate information about the effectiveness and benefits of MDA to promote acceptability and uptake among school children.

Primary school children who understood the purpose of MDA were more likely to have a high acceptability of MDA. This could indicate that when children understand the purpose of MDA, they may perceive it as a valuable tool for disease prevention rather than an arbitrary requirement. This understanding likely instills a sense of trust and confidence in the intervention, leading to greater acceptance and willingness to participate. In line with this finding, a study conducted by Tuhebwe *et al* [18], reported that insufficient knowledge of schistosomiasis prevention results in children's unwillingness to take the praziquantel tablets. Additionally, another study conducted in Nigeria also found that a lack of sufficient knowledge about the diseases and control intervention acted as a barrier to uptake and willingness to participate in the intervention [32]. The strong association between understanding of the intervention purpose and acceptability of MDA in this study highlights the need to provide adequate information about MDA to school children to enhance their understanding of its purpose.

**Table 2. Factors associated with acceptability of schistosomiasis MDA (Unadjusted and adjusted logistic regression results).**

| Variable | Unadjusted | | | Adjusted | | |
|---|---|---|---|---|---|---|
| | COR | 95% CI | p-value | AOR | 95% CI | p-value |
| **Age (in years)** | 1.21 | 1.06–1.37 | 0.004* | – | – | – |
| **Gender** | | | | | | |
| Girls | Ref | | | | | |
| Boys | 0.94 | 0.68–1.29 | 0.685 | 0.59 | 0.38-0.94 | **0.027*** |
| **Understanding of intervention purpose** | | | | | | |
| No | Ref | | | | | |
| Yes | 14.55 | 9.52–22.23 | <0.0001* | 5.51 | 3.16-9.59 | **<0.0001*** |
| **Have ever swallowed praziquantel** | | | | | | |
| No | Ref | | | | | |
| Yes | 11.14 | 6.16–20.14 | <0.0001* | – | – | – |
| **Affective attitude** | | | | | | |
| Uncomfortable | Ref | | | | | |
| Comfortable | 17.38 | 11.19–26.99 | <0.0001* | 5.10 | 2.77-9.39 | **<0.0001*** |
| **Perceived effectiveness** | | | | | | |
| No | Ref | | | | | |
| Yes | 15.30 | 9.27-25.26 | <0.0001* | 2.52 | 1.31- 4.85 | **0.006*** |
| **Waiting time** | | | | | | |
| Short wait time | Ref | | | | | |
| Long wait time | 0.81 | 0.26–2.53 | 0.711 | – | – | – |
| **Self-efficacy** | | | | | | |
| Unconfident | Ref | 9.05–21.51 | <0.0001* | 2.04 | 1.08-3.85 | **0.029*** |
| Confident | 13.95 | | | | | |
| **Perceived moral about taking praziquantel** | | | | | | |
| No | Ref | | | | | |
| Yes | 0.52 | 0.29–0.93 | 0.026* | – | – | – |
| **Interference with school activities** | | | | | | |
| No | Ref | | | | | |
| Yes | 1.53 | 1.06–2.21 | 0.021* | – | – | – |

*Statistically significant results, COR = Crude odds ratio, AOR = Adjusted odds ratio, CI = Confidence interval.

The study also showed that being confident in taking schistosomiasis drugs was positively associated with high acceptability. One possible explanation for this could be those who were more confident in taking schistosomiasis drugs they might have received thorough education or information about the benefits of the medication, its mode of action, and potential side effects. This knowledge equips them with the confidence to overcome any challenges associated with the treatment. Similar findings are reported in other studies conducted in Northern Samar, Philippines, and Zimbabwe [26,36], which documented that pupils' empowerment and awareness regarding schistosomiasis MDA improved their participation in such programs. This suggests the importance of encouraging and empowering children to understand the rationale behind MDA. Furthermore, the study found that affective attitude was associated with acceptability of MDA. It seems like individuals' emotional responses and feelings towards the intervention influence their willingness to participate. When individuals have a positive attitude towards MDA, such as feeling comfortable, reassured, or optimistic about the intervention, they are more likely to accept it. This finding is consistent with another study conducted in Uganda, which found that most children reported experiencing discomfort for several days after taking medication as a barrier to participating in MDA [37].

This result indicates that addressing emotional responses through tailored communication and community engagement could enhance MDA acceptability and uptake.

Additionally, male primary school children were less likely to have high acceptability of MDA compared to females. The probable reason for this gender disparities might indirectly relate to concerns about reducing male potency. In some cultures, or communities, there could be traditional beliefs or misconceptions surrounding health interventions, where participating in activities like MDA might be perceived as weakening or diminishing one's masculinity or male potency. Perhaps this could also be the case in the Tanzania setting as similar concerns also occurred during the distribution of mosquito bed net. Consistent with this finding, other studies conducted in KwaZulu-Natal, South Africa and Mpigi district, Uganda found that males were less likely to adhere to MDA compared to females [38,39]. However, these results contradict the findings of a study in rural Philippines, which revealed that females showed a higher risk of non-compliance due to reasons such as breastfeeding or pregnancy at the time of MDA [19]. The observed differences in study outcomes may be attributed to variations in the study population. Unlike the study conducted in rural Philippines that focused on adults, our investigation was conducted among primary school children. These demographic differences likely introduce varying behavioral patterns, environmental exposures, and socioeconomic factors, which can influence study outcomes. Nevertheless, other studies have found that sex is not a significant determinant of compliance and willingness to participate in MDA [40], indicating the need for further investigation into the multifaceted determinants of treatment acceptability in diverse contexts.

## Strengths and limitation of the study

The study utilized a validated tool for data collection, ensuring the reliability and validity of the gathered data. Additionally, the questionnaire was developed based on well-established concepts of acceptability that have been utilized in previous studies, providing a strong theoretical foundation. This instills confidence in the quality and accuracy of the collected data, thereby enhancing the robustness of our findings.

One potential limitation of this study pertained to self-reported data on acceptability; some respondents were unable to recall treatment correctly, as there are usually two campaigns: one for albendazole to treat soil-transmitted helminths and another for praziquantel to treat schistosomiasis. To address this, research assistants provided visual demonstrations of both praziquantel and albendazole tablets during the interviews, ensuring that participants fully understood the interventions of this study. This proactive approach significantly minimized issues related to recall errors, enhancing our confidence in the accuracy of the self-reported data within this study.

## Conclusion

This study highlighted several critical findings that inform the acceptability of Mass Drug Administration (MDA) among primary school-aged children in rural Tanzania. Slightly more than half of the primary school children scored high acceptability of MDA. However, variables such as perceived effectiveness, understanding of the intervention's purpose, self-efficacy, affective attitude, and gender strongly influence acceptability. Furthermore, gender disparities were observed, with male children showing lower acceptability. This suggests a need for tailored, gender-sensitive community sensitization efforts regarding the benefits of MDA. By focusing on factors influencing the acceptability of MDA and implementing gender-sensitive strategies, we can achieve higher treatment coverage, ultimately mitigating the burden of schistosomiasis in affected communities. Implementers should build on these insights in refining MDA campaign strategies, emphasizing the importance of context-specific approaches in the effective control and prevention of neglected tropical diseases.

## Supporting information

**S1 File. This is the questionnaire in Swahili.** This file contains the complete version of the study questionnaire translated into Swahili, which was used for data collection.
(DOCX)

**S2 File. This is the questionnaire in English.** This file provides the original English version of the questionnaire used in the study.
(DOCX)

**S3 File. Inclusivity in global health questionnaire.** This file contains the completed version of PLOS' Inclusivity in Global Research questionnaire, submitted in accordance with the journal's policy to improve transparency in studies conducted outside the researchers' own country or Community.
(DOCX)

**S1 Data. The data set used for analysis.** This Excel file contains the anonymized raw data collected from the questionnaires, which was used for all statistical analyses in the manuscript.
(XLSX)

## Acknowledgments

Irene Yunzu Edward is a recipient of a TDR scholarship under the postgraduate training scheme in implementation research at the University of Zambia, School of Public Health. We are grateful for the support from the training scheme, as provided by the UNICEF/UNDP/World Bank/WHO special programme for Research and Training in Tropical Diseases (TDR).

## Author contributions

**Conceptualization:** Irene Yunzu Edward, Patricia Maritim, Choolwe Jacobs, Hikabasa Halwiindi.

**Data curation:** Irene Yunzu Edward.

**Formal analysis:** Irene Yunzu Edward.

**Investigation:** Irene Yunzu Edward.

**Methodology:** Irene Yunzu Edward, Patricia Maritim, Choolwe Jacobs, Hikabasa Halwiindi.

**Software:** Irene Yunzu Edward.

**Supervision:** Patricia Maritim, Choolwe Jacobs, Hussein Mohamed, Hikabasa Halwiindi.

**Visualization:** Irene Yunzu Edward.

**Writing – original draft:** Irene Yunzu Edward.

**Writing – review & editing:** Irene Yunzu Edward, Patricia Maritim, Choolwe Jacobs, Adam Silumbwe, Hussein Mohamed, Joseph Mumba Zulu, Hikabasa Halwiindi.

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
