## [Decision Letter · Decision Letter 0]

Dear Dr. Edward,

Thank you for submitting your manuscript to PLOS ONE. After careful consideration, we feel that it has merit but does not fully meet PLOS ONE’s publication criteria as it currently stands. Therefore, we invite you to submit a revised version of the manuscript that addresses the points raised during the review process.

**ACADEMIC EDITOR:**

The two reviewers have made some suggestions for improvement of the manuscript. Overall, these are minor edits and should help to improve the manuscript.The manuscript has merit and is well written. I suggest a minor edit under the Methods on theoretical framework with respect to the definition of intervention coherence- check the grammar and revise.  

We look forward to receiving your revised manuscript.

Kind regards,

Kenneth Bentum Otabil, PhD

Academic Editor

PLOS ONE

Journal Requirements:

Additional Editor Comments:

The study is titled 'Determinants of acceptability of schistosomiasis mass drug administration among primary school children in Busega District, Northwestern Tanzania' and sought to establish factors associated with acceptability of schistosomiasis mass drug administration among primary school children in Busega District. The objectives, methods, results and discussions are sufficiently done but kindly take care to revise the the manuscript based on the comments of the reviewers.

One other very minor correction: Under the theoretical framework, the sentence 'Intervention coherence related to the extent to which the participant understands the purposes of MDA' seems incomplete. Kindly revise. Consider 'Intervention coherence is....................................

Reviewers' comments:

Reviewer's Responses to Questions

**Comments to the Author**

1. Is the manuscript technically sound, and do the data support the conclusions?

Reviewer #1: Yes

Reviewer #2: Yes

2. Has the statistical analysis been performed appropriately and rigorously?

Reviewer #1: Yes

Reviewer #2: Yes

3. Have the authors made all data underlying the findings in their manuscript fully available?

Reviewer #1: Yes

Reviewer #2: Yes

4. Is the manuscript presented in an intelligible fashion and written in standard English?

Reviewer #1: Yes

Reviewer #2: Yes

Reviewer #1: This is an interesting social study on Praziquantel MDA.

I recommend authors to add more details on how the children consented, whether it was individually or not.

Additionally, since MDA is done to all SAC i.e. 5 years and above (till end of Primary school), there is a need to explain why authors chose to include only those in grade V-VII in the study.

Add when MDA (year) in reference was conducted? I just see that data collection time March- May 2023.The gap between the two timelines could contribute to some variables in question.

Reviewer #2: Review of Manuscript

Title: Determinants of acceptability of schistosomiasis mass drug administration among primary school children in Busega District, Northwestern Tanzania.

Introduction: Kindly rephrase this statement for clarity “Acceptability is a multi-faceted construct that reflects the extent to which receiving an intervention consider it to be appropriate, based on anticipated or experienced cognitive and emotional responses to the intervention”

1. Is the manuscript technically sound, and does the data support the conclusions?

Answer: The study ‘Determinants of acceptability of schistosomiasis mass drug administration among primary school children in Busega District, Northwestern Tanzania’aimed at assessing factors associated with acceptability of schistosomiasis MDA among school children in Busega District, North-western Tanzania was technically sound. And the data obtained supports the conclusions of the study.

2. Has the statistical analysis been performed appropriately and rigorously?

Answer: Statistical analysis was appropriately and rigorously done as primary data collected were imported, cleaned, coded and analyzed using the statistical software STATA version 15 (STATA corp. college station, Taxes, US). The Cronbach Alpha which measures the internal consistency (the extent to which the items in a test measure the same component) of a scale was used. The analysis performed was appropriate and rigorous which produced good results.

3. Have the authors made all data underlying the findings in their manuscript fully available?

Answer: All data were provided in relations to this work. And the ethical consideration was taken which was necessary for the study to be conducted.

4. Is the manuscript presented in an intelligible fashion and written in standard English?

Answer: The manuscript follows the format for PLOS ONE with the language written well which is easy to understand. I recommend this manuscript for publication however, correction should be done for the introduction part for clarity.

**Do you want your identity to be public for this peer review?** For information about this choice, including consent withdrawal, please see our Privacy Policy

Reviewer #1: No

Reviewer #2: **Yes: ** Anabel Acheampong

---

## [Author Response · Author response to Decision Letter 1]

13 Jun 2025

ACADEMIC EDITOR

1. The two reviewers have made some suggestions for improvement of the manuscript. Overall, these are minor edits and should help to improve the manuscript.

Thanks for taking time to read the manuscript and sharing your valuable comments.

2. The manuscript has merit and is well written. I suggest a minor edit under the Methods on theoretical framework with respect to the definition of intervention coherence- check the grammar and revise.

Thank you for the comment. We have revised the definition for clarity. It now reads: Intervention coherence is defined as the extent to which participants understand the purpose of the intervention (MDA) and how it works (Page 5, lines 130–132).

Additional Editor Comments

The study is titled 'Determinants of acceptability of schistosomiasis mass drug administration among primary school children in Busega District, Northwestern Tanzania' and sought to establish factors associated with acceptability of schistosomiasis mass drug administration among primary school children in Busega District. The objectives, methods, results and discussions are sufficiently done but kindly take care to revise the the manuscript based on the comments of the reviewers.

Thanks for taking time to read the manuscript and sharing your valuable comments

One other very minor correction: Under the theoretical framework, the sentence 'Intervention coherence related to the extent to which the participant understands the purposes of MDA' seems incomplete. Kindly revise. Consider 'Intervention coherence is....................................

Thank you for the comment. We have revised the definition for clarity. It now reads: Intervention coherence is defined as the extent to which participants understand the purpose of the intervention (MDA) and how it works (Page 5, lines 130–132).

REVIEWERS' COMMENTS:

REVIEWER REPORTS

Reviewer #1:

1.This is an interesting social study on Praziquantel MDA.

I recommend authors to add more details on how the children consented, whether it was individually or not.

Thank you for the comment. We have added this information in the ethical considerations section (Pages 9, lines 258–260).

Assent forms were provided to eligible primary school children. The consent process was conducted individually, whereby each child who agreed to participate signed an assent form.

2. Additionally, since MDA is done to all SAC i.e. 5 years and above (till end of Primary school), there is a need to explain why authors chose to include only those in grade V-VII in the study.

Thank you for the comment. Although MDA targets all school-age children (SAC) aged 5 years and above, this study focused on students in grades V–VII for several considerations. First, these students had the full benefit of two to three rounds of MDA (2018, 2019, and 2021), providing sufficient exposure to evaluate their experiences. In addition, older students are more likely to comprehend and articulate their experiences, perceptions, and concerns regarding the intervention, which is critical in assessing acceptability. This approach also aligns with similar studies on MDA, which have prioritized upper primary students (26,27).

We have explained this in detail in study population section (Page 6, lines 166-175).

3. Add when MDA (year) in reference was conducted? I just see that data collection time March- May 2023.The gap between the two timelines could contribute to some variables in question.

Thank you for the comment. The last round, which MDA was conducted in this district was 2021. No mass praziquantel treatment was administered in 2022, a year before data collection for this study was done (Page 5, lines 147-148).

Reviewer #2:

1. Title: Determinants of acceptability of schistosomiasis mass drug administration among primary school children in Busega District, Northwestern Tanzania.

Introduction: Kindly rephrase this statement for clarity “Acceptability is a multi-faceted construct that reflects the extent to which receiving an intervention consider it to be appropriate, based on anticipated or experienced cognitive and emotional responses to the intervention.

Thank you for the comment. We have revised the definition to make it clearer. It now reads: Acceptability is the perception among implementation stakeholders that a given treatment, service, practice or innovation is agreeable, palatable, or satisfactory (Page 4, lines 105-107).

---

## [Editor Report · Decision Letter 1]

Determinants of acceptability of schistosomiasis mass drug administration among primary school children in Busega District, Northwestern Tanzania

PONE-D-25-05835R1

Dear Dr. Edward,

We’re pleased to inform you that your manuscript has been judged scientifically suitable for publication and will be formally accepted for publication once it meets all outstanding technical requirements.

Kind regards,

Kenneth Bentum Otabil, PhD, PhD

Academic Editor

PLOS ONE
---

## [Editor Report · Acceptance letter]

PONE-D-25-05835R1

PLOS ONE

Dear Dr. Edward,

I'm pleased to inform you that your manuscript has been deemed suitable for publication in PLOS ONE. Congratulations! Your manuscript is now being handed over to our production team.

Kind regards,

on behalf of

Dr. Kenneth Bentum Otabil

Academic Editor

PLOS ONE